# Treatments and Prognosis for Subchondral Cystic Lesions in the Distal Extremities in Thoroughbred Prospect Racehorses

**DOI:** 10.3390/ani13182838

**Published:** 2023-09-07

**Authors:** Marcos Pérez-Nogués, Javier López-Sanromán, Michael Spirito, Gabriel Manso-Díaz

**Affiliations:** 1Peterson & Smith Equine Hospital, 4747 SW 60Th St., Ocala, FL 34474, USA; 2Department of Animal Medicine and Surgery, Universidad Complutense de Madrid, Avenida Puerta de Hierro s/n, 28040 Madrid, Spaingmanso@ucm.es (G.M.-D.); 3Hagyard Equine Medical Institute, 4250 Iron Works Pike, Lexington, KY 40511, USA; mspirito@hagyard.com

**Keywords:** horse, osteochondrosis, sales, subchondral cystic lesion, orthopedics

## Abstract

**Simple Summary:**

Subchondral cystic lesions (SCLs) are spheric/oval-shaped demineralized cavities that develop in equine bones at a young age (3–18 months of age). SCLs are a form of developmental bone disease, and, while their etiology is incompletely understood, they may be a manifestation of osteochondrosis (OC). This subchondral bone absence in SCLs impacts the distribution of weight-bearing loads, which can result in lameness and poor performance in racehorses. Also, the communication between SCLs and the affected joint can create a step defect in the cartilage, which may trigger arthritis. Several surgical and medical treatments have been developed to improve the athletic prognosis of horses with SCLs. We compared four treatments for SCLs in the metacarpal, metatarsal, and phalangeal bones of young prospect racehorses and found that all treatments have similar prognoses. We also found that horses treated for SCLs have similar racing careers compared to those of their siblings, despite having a difference in purchase price as yearlings. Horses with higher and wider SCLs were found to have fewer wins and places during their racing careers, indicating that SCLs negatively affect the quality of racing.

**Abstract:**

Subchondral cystic lesions (SCLs) in equines and their treatments have been mainly studied in the medial femoral condyle of the femur. SCLs in the distal extremities affecting the fetlock or interphalangeal joints are frequent, but treatment or prognosis studies in horses are currently sparse. Our objective was to compare four treatments for SCLs in the distal extremities (intralesional injection of corticosteroids, transcortical drilling, cortical screw placement, and absorbable hydroxyapatite implant placement) and report the racing prognoses for affected thoroughbred yearlings. Data from 113 thoroughbred yearlings treated for SCLs in the distal extremities were collected from 2014 to 2020. Age at surgery, sex, bone affected, radiographic SCL measurements, SCL shape, and type of treatment were recorded. Sale data and racing performance were collected for the operated horses and for 109 maternal siblings that were free of SCLs. An analysis was conducted to assess if SCL size affected racing prognosis and to detect differences in sale value and selected racing parameters between the cases and controls. The outcomes for the different treatments, the different bones affected, and the SCL shape type were also analyzed. There was no difference in the ability to start in a race between the cases and controls (60.2% vs. 69.7%, respectively). The auction value of the treated horses was significantly lower than that of their siblings. The bone affected did not impact any of the racing variables studied, whereas the height of the SCLs negatively affected the number of wins and placed races. The type of treatment for the horses affected by SCLs did not have an impact on sale prices, ability to start a race, race starts, wins, and places, or age at the time of the first start. In conclusion, yearlings diagnosed with an SCL in the distal extremities had lower auction prices and decreased racing performances, with lower numbers of wins correlated with larger SCL heights compared to the siblings. Similar racing performance was found regardless of the treatment received.

## 1. Introduction

Subchondral cystic lesions (SCLs) in horses have been extensively studied. The most common location for SCLs is the medial condyle of the femur. The most frequent areas affected by SCLs in the distal extremities of horses are the condyles of the distal third metacarpus and metatarsus, while SCLs in the various phalanges are relatively uncommon [1].

Most SCLs develop in weanlings and yearlings, and clinical lameness is not always present on soundness evaluation. Radiographic monitoring at an early age could help categorize which horses need surgical or medical intervention [2]. SCLs with a focal, narrow communication into the synovial cavity theoretically could respond better to more conservative approaches, as they have less articular cartilage surface affected, and certain treatments could have different results depending on the shape and size of the lesions [3,4,5]. Based on the studied morphology of SCLs, their center is composed of fibrous tissue rich in fibroblasts with upregulated interleukin-6 and interleukin-1β expression mRNA [6]. This fibrous tissue is thought to be one of the main causes of bone resorption in SCL development and might be responsible for the persistence or expansion of the lesion over time [6,7].

The scientific literature cites numerous medical and surgical options for SCLs. Medical treatments reported within the literature include rest, intraarticular or SCL intralesional injection with corticosteroids, plasma-derived products, and stem cells [8,9]. Surgical treatments such as bone filling with cancellous bone grafts, arthroscopic or external debridement (transcortical drilling), stainless steel screw or composite absorbable transcondylar implants, cartilage grafts, and mosaic arthroplasty have also been described [10,11,12,13,14,15,16,17,18,19]. Because the most prevalent location is the medial femoral condyle, most of the treatment options have been studied in that location. Similar soundness prognoses were reported in retrospective publications after arthroscopic debridement (56–64%), arthroscopic-guided intralesional corticosteroid injection (67%), and cancellous bone graft after debridement (50%) [11,17,18]. Allogenic chondrocyte grafting with IGF-1 and transcondylar cortical screw placement showed slightly better soundness success (74 and 75%, respectively) [12,17]. A recent report using a composite bioabsorbable implant reported that 94% of horses returned to soundness after surgery based on a referring veterinarian lameness recheck evaluation, and 71% of horses returned to racing after surgery [16].

Few reports examining the treatment and prognosis of phalangeal, metacarpal, or metatarsal SCLs have been published in equines [20,21,22]. In a retrospective case series, horses with treated SCLs of the distal extremities had a better prognosis for returning to soundness and greater radiographic evidence of SCL filling with bone than horses with medial femoral condyle lesions. A total of 80% of horses were sound after arthroscopic debridement of a third metacarpal SCL, as were 91% of horses with an SCL in the distal phalanx [20,21]. However, a negative surgical result was seen in a one-case report of an SCL in the middle phalanx [22].

The prognosis of distal extremity SCLs in equines is infrequently reported, and there are no current studies comparing different treatments for SCLs in the distal extremities [1]. The treatments currently used in our institution and described in this report are as follows: intralesional corticosteroid injection, extraarticular debridement by drilling of the SCL, translesional cortical screw placement in lag fashion, and absorbable biocomposite implant placement. The objective of this study was to compare the athletic prognosis of a population of racing prospect thoroughbred yearlings that underwent different treatments for metacarpal/metatarsal and phalangeal SCLs and compare their performances as racehorses to the performance of their maternal siblings. We hypothesized that yearlings with radiographic evidence of SCLs in the bones of interest will have a lower sale price at an auction, poorer performance in terms of earnings, wins, and placed races, and less active athletic careers with fewer starts. We also hypothesized that there would be no difference in performance outcomes between treatment groups.

## 2. Materials and Methods

### 2.1. Data Collection

Data were collected from all thoroughbred yearlings (less than 24 months of age) that underwent an intralesional corticosteroid injection, extraarticular debridement by drilling, translesional cortical screw placement in lag fashion, or absorbable biocomposite implant placement to treat SCLs in the distal extremities from 2014 to 2020 at a referral hospital in Kentucky, USA. Data collected included age in days at the time of the procedure, sex, limb, bone affected, type of treatment, and lesion size. Auction and racing data were collected from a public database (www.equibase.com, accessed on 1 August 2023) and included performance (number of starts, wins, and placed races), total earnings in US dollars, age in days at the time of the horse’s first race until 31 July 2023, and sale purchase price. The control group were maternal siblings that were born a year prior (or a year later if the mare did not have a foal the year prior) and had no clinical history of presentation for SCLs.

### 2.2. Image Analysis

Preoperative radiographs in DICOM format were evaluated blindly in a randomized order by three observers using medical imaging viewer software (OsiriX^®^ Lite v12.0.1 DICOM viewer, Pixmeo SARL, Bernex, Switzerland). The width and height of the lesions were measured in dorsopalmar/plantar radiographic views. The magnification was corrected by calculating the index of the measured SCL with the total width of the specific joint affected. Each measurement was taken once by three authors. The mean of the three different measurements was calculated and used for statistical purposes. SCL shape was classified in three categories, following previous classification guidelines (Table 1 and Table 2) [8]. The intraclass correlation coefficient was calculated to make sure that there was a high level of agreement between observers when taking height and width measurements and when classifying SCLs by shape.

### 2.3. Treatments

Horses were classified into 4 groups based on the type of procedure performed. Patients were not pre-randomized into treatment groups, but each attending surgeon performed just one of the described treatments for their patients, regardless of SCL size, shape of the lesion, or owner expectations. The treatment groups were as follows: SCL injection with corticosteroids, SCL drilling, cortical screw placement, and absorbable hydroxyapatite implant placement. All treatments were performed before the horses were attempted to be sold in an auction. Briefly, a corticosteroid injection was performed under general anesthesia with the horse placed in dorsal recumbency, and the affected joint was flexed. The site of injection was determined with ultrasound, and, after clipping and sterile preparation of the area, the SCL was injected with 40–80 mg of methylprednisolone acetate (Depo-Medrol; 20 mg/mL; Zoetis, Kalamazoo, MI, USA) or 12 mg of betamethasone acetate (BetaVet; 6 mg/mL; American Regent, Inc., Animal Health, Shirley, NY, USA) at several spots of the SCL lining. After the corticosteroid injection, 3 mL of a glycosaminoglycan suspension (Polyglycan; 10 mL vial; Bimeda, Oakbrook Terrace, IL, USA) mixed with 125 mg of amikacin (amikacin sulfate injection; 500 mg/2 mL; Avet Pharma, East Brunswick, NJ, USA) was injected in the joint. To perform the transcortical approach to debride the SCL using a drill bit, the patient was positioned in lateral recumbency with the SCL lesion side uppermost. The size of the drill bit was selected according to the size of the cystic cavity, and the direction taken through the cortical bone was guided by digital radiography. A similar technique was published in a case report but was CT-guided [22]. The cortical screws and absorbable implants were placed as previously described [12,16]. In brief, the horse was anesthetized and placed in lateral recumbency with the lesion uppermost. A correct drill bit size was used for the selected implant, and the hole was drilled starting in the cortex closest to the lesion. The drilling direction was guided radiographically across the bone parallel to the joint surface and traveled through the SCL. Cortical screws were placed in lag fashion across the SCL, whereas absorbable implants were placed into the SCL in a neutral position.

### 2.4. Statistical and Data Analysis

Data were analyzed with the SAS statistical analysis software, version 9.4 (SAS Statistical Software 9.4; Cary, NC, USA). All numerical data did not show a normal distribution when the Shapiro–Wilk test was used. A Wilcoxon rank sum test comparing the cases and controls was performed to detect differences in sale value, number of starts, wins, and places, and day of the first race. Pearson’s chi square test was used to detect any effects of the location of the SCL on the chances of the horse starting a race and compare sex and the ability to start at least one race between the cases and controls. A Kruskall–Wallis equality-of-population test was performed to detect differences in sale value, earnings, age at first race, age at surgery, number of wins, number of places between types of SCL, type of surgery performed, and SCL location. A Spearman correlation was used to examine if the height or width of the SCL had any association with wins or the number of races placed. For all tests, a *p* ≤ 0.05 was considered significant.

A multivariate negative binomial regression analysis was conducted to determine if sex, age at surgery, SCL height, SCL width, SCL shape, bone affected, type of surgery, front or back leg affected, and lateral or medial location inside the joint could predict the number of starts, number of wins, and number of placed races. Full model building was performed, and the criterion for the removal of the dependent variable was a *p* > 0.05. A linear regression was conducted to detect any association between total career earnings and the predictive variables height, width, shape, bone affected, type of surgery, yearling auction price, front or back leg, and medial or lateral location. A negative binomial regression was also conducted to detect if having the horse undergo surgery at a younger age would have any predictable association with the age at first race, earnings, wins, number of races placed, and starts. A binary logistic regression was conducted to determine if height, width, shape, bone affected, type of surgery, auction price, front or back leg, and medial or lateral location could predict the ability of a patient to start or not start a race.

## 3. Results

A total of 121 SCLs in 113 horses were treated during the study. Group classification and performance parameter results are compiled in Figure 1 and Table 2 and Table 3. Of those, 72 horses had SCLs located distally in the third metacarpal or metatarsal bone, 32 in the first phalanx, and 9 in the middle phalanx. The control group consisted of 109 maternal siblings, as 4 horses did not have a sibling in the prior or following year. In the study group, a similar number of males (n = 60; 53.1%) and females (n = 53; 46.9%) were found, which was not different from the control group (males n = 53; 48.6%; females n = 56; 51.4%). The SCLs were located in the forelimbs of 84 horses (74.3%) and in the medial aspect of the joints (n = 86; 71.1%). Six horses were diagnosed with SCLs affecting two joints (two horses had SCLs in both front distal metacarpal bones; one had SCLs in both metatarsal bones affected; one horse had SCLs in one metacarpal and one metatarsal bone; one horse had SCLs in one metacarpal bone and in the proximal phalanx of the same leg; and one horse had an affected middle phalanx in the bilateral forelimbs); and one horse had three joints affected (the SCLs were in both distal metacarpal bones and in the proximal phalanx in the hind leg). The frequency of affected left versus right limbs was similar (54.8% and 45.2%, respectively). At the final date of performance data collection, nine cases (8%) and six controls (5.5%) were still actively competing in flat racing.

The median and interquartile ratios of the SCL height and width were 9.47 mm (6.97–10.92 mm) and 8.58 mm (6.98–10.96 mm), respectively. The SCLs located in the distal metacarpal/metatarsal bones measured 9.03 mm (6.96–10.9 mm) in height and 8.52 mm (7–10.92 mm) in width. A more oval shape was found in the lesions of the proximal phalanx, with a median height of 10.8 mm (7.03–10.95 mm) and a median width of 7.78 mm (7.09–10.96 mm). A rounder shape was found in the distal phalanx, with a median of 10.28 mm (7.05–10.97 mm) for height and 10.81 mm (7.03–10.95 mm) for width. The different bones affected by the SCLs were no different in the horse’s ability to start a race (*p* = 0.33).

In 61 horses, the SCL was injected with corticosteroids; 30 horses underwent transcortical drilling of the SCL; 16 had a cortical screw placed; and 6 had a hydroxyapatite-based absorbable implant placed. The mean age to perform any of the treatments was 373 ± 125 days (13 months), and the mean age of the first start was 1045 ± 262 days (2.8 years). These values were similar between the affected horses and controls (1045 days vs. 1023 days, respectively; *p* = 0.44). A younger age of the horse at the time of surgery was predictive of a younger age at first race (IRR = 1.01 (1–1.01) (*p* = 0.02). The ability to start a racing career was not significantly different among the types of procedures performed (*p* = 0.38). Also, the type of surgery did not have an impact on the sale prices (*p* = 0.33), earnings (*p* = 0.68), wins (*p* = 0.36), placed races (*p* = 0.28), age at the time of surgery (*p* = 0.54), or age at the first race in horses operated for SCLs (*p* = 0.99). Horses diagnosed with multiple SCLs had similar prognoses for starting racing careers compared to horses with just one SCL (*p* = 0.02).

The intraclass correlation coefficient confirmed a high interrater reliability for height measurement (ICC = 0.925 [95% C.I. 0.893–0.949]), width measurement (ICC = 0.876 [95% C.I. 0.822–0.915]), and shape classification of the SCLs (0.795 [95% C.I. 0.706–0.86]).

There was no difference (*p* = 0.12) in the percentage of horses sold at a yearling auction between the cases (64.6%) and controls (73.4%). There was no difference between the cases and controls in the ability to start at least one race (n = 68, 60.2% vs. n = 76, 69.7%, respectively; *p* = 0.1). The sale price of the horses with an SCL was lower than their siblings (*p* < 0.01) (Table 4). The total earnings were lower but statistically similar in horses with SCLs than in controls (*p* = 0.1) (Table 4). The number of races won and placed did not differ for the cases and controls, and the age in days at which a horse debuted in a race did not differ either (Table 4).

The negative binomial regression model to predict the number of wins, placed, and starts from SCL variables (height, width, type of SCL, age at surgery, age at first start, treatment, bone affected, front or back leg, and medial or lateral) was significant (Model Wins X^2^(9) = 18.78, *p* = 0.03; Model Placed X^2^(8) = 16.29, *p* = 0.04; Model Starts X^2^(9) = 21.71, *p* = 0.01). The variable SCL height and the age at surgery were significant negative predictors of the number of wins (height IRR = 0.896 [95% C.I. = 0.737–0.943]; *p* < 0.01; age at surgery IRR = 0.997 [95% C.I. = 0.994–0.999]; *p* < 0.01). The variable age at the time of surgery was a predictor variable of the number of races placed (IRR = 0.997 [95% C.I. = 0.995–0.999]; *p* = 0.01) and the number of starts (IRR = 0.998 [95% C.I. = 0.996–0.999]; *p* < 0.01). The linear regression model to predict total earnings was not statistically significant (R^2^ = 0.26, F(8,34) = 1.492, *p* = 0.2), but the auction price was a moderately significant predictor for higher total career earnings (β = 0.506, t(99) = 2.982, *p* < 0.01). The binary logistic regression model to predict the ability to race based on the SCL characteristics and auction value was significant (X^2^(9) = 18.08; *p* = 0.04), and it showed that the medial location of the SCL in the joint was a significant predictor of a decreased ability to start a race (OR = 4.6 [95% C.I. = 1.23–17.235]; *p* = 0.02). The Spearman correlation test showed a significant negative correlation between SCL height and number of wins (r [57] = −0.295; p = 0.03) and number of placed races (r [57] = −0.263; p = 0.04). Additionally, it also showed a negative correlation between SCL width and placed races (r [57] = −0.261; p = 0.05).

## 4. Discussion

We found that subchondral cystic lesions in the third metacarpal/metatarsal bones or in the distal phalanges had a negative impact on the price of the yearlings at an auction but did not have a significant impact on starting a career as a racehorse when compared to siblings. However, SCLs of greater height were associated with a decrease in the number of won and placed races. Also, SCLs located on the medial side of the joint had a greater negative impact on the ability to start in at least one race than lateral SCLs.

The purchase values achieved at auctions were lower for horses with SCLs than those of their siblings. This was expected and found in similar studies reporting on radiographic lucencies in the medial femoral condyle larger than 4mm in height [2,23]. Also, a higher yearling sale price was a predictive value of more earnings as a racehorse, which, to our knowledge, has not been reported in previous racehorse publications. We chose to analyze several routinely studied performance parameters based on a previous meta-analysis [24]. The linear regression model for earnings was not statistically significant, so the combination of the SCL morphology and size variables was not able to predict this performance outcome. Contrary to our hypothesis, the ability of a yearling to start a race was not affected by the presence of an SCL or the applied treatments in these locations. The age at the first race was not different between the cases and controls, but a later treatment intervention significantly delayed the first race. Thus, earlier treatment might be preferred in some cases and could be beneficial if a second intervention is desired or needed. Furthermore, SCL enlargement is a possibility, and early intervention if SCL expansion is detected in sequential radiographs has been recommended [2,15].

The forelimbs and medial aspects of the joints were more commonly affected, which was also seen in other publications [1,20,22]. As the medial metacarpal and metatarsal condyles are wider, they sustain higher loads than the lateral condyles [1]. This probably influences the higher medial location prevalence and may be the reason why we found a predictive association for a horse’s decreased ability to start a race when the medial side of the joint is affected. Several studies have tried to correlate the size and shape of SCLs with racing career outcomes in the medial femoral condyle, but the results were contradictory [3,4,8,12,15,18]. Small concavities < 3 mm deep in the medial femoral condyle did not have an effect on performance, whereas larger SCLs and dome shapes have been reported to have worse prognoses for racing [2]. We failed to detect any difference in performance based on our shape classification of the SCLs (Table 1). This classification was modified from previous models and was based on the shape of the SCL at the articular surface [4,8]. A larger articular communication has been suggested to reduce a horse’s ability to start a race by increasing the detrimental effects on the cartilage [3,25]. The height of the SCL had the strongest association with decreased performance. SCLs with greater height have a reduced amount of compact subchondral bone, which may affect the weight-bearing load distribution within the bone. The uneven weight load forces when there is a void in the subchondral bone have been modeled in the medial femoral condyle [5]. Different bone-adapting changes can be expected when the rest of the subchondral bone surrounding the SCL supports the entire load-bearing force in the stifle and in all other joints. This was modeled to be more pronounced as the SCL height increases [5]. The adapting or maladapting of bone remodeling may be associated with pain during bone deposition or resorption and may be the reason for lameness or poor performance [4,5].

The auction value of the horses after the surgery was similar among the different treatment groups, which may confirm that the potential value of the horses at the time of surgery did not affect treatment decisions. All the treatments were performed within the same period, and a low percentage of horses were still actively racing in both the treatment and control groups, which may have some repercussions in the analyses. However, because of the low number of actively racing horses in both groups, we expected minimal repercussions in the data. None of the different treatment methods used (with different therapeutic targets) had a significantly better long-term outcome. Because we did not follow up on the defect filling with new bone radiographically, it is unknown if the surgical removal of the fibroblasts and endothelial cells from the center of the SCL could have any improved benefits in radiographic resolution compared to intralesional medication. The limited flexion capacity of the pastern joint makes access for intralesional injection difficult. Thus, some of the middle and distal proximal phalanx SCLs may only be able to be treated by drilling or implant placement. However, in this study, we were able to effectively inject type 2 and 3 SCLs located in the pastern joint by placing the needle under radiographical guidance.

Recently, the use of a composite bioabsorbable implant reported a better outcome for soundness than other techniques applied to the medial femoral condyle. However, this report used referring veterinarians to determine success [16]. Only 48% of weanlings and yearlings raced as 2-year-olds in Europe after bioabsorbable implant placement in the medial femoral condyle, and, unfortunately, these horses were not compared to a control group without pathology. This bioabsorbable implant technique has been reported to have the added benefit of improving the radiographic appearance [16]. However, data reported in a recent study showed that radiographic improvement after arthroscopic debridement of SCLs in the medial femoral condyle did not correlate with soundness in a small population sample [15]. Although the bioabsorbable implant group was the smallest in our population, the horses in this group did not perform significantly better than any of the other horses that underwent other SCL treatments in the distal extremity bones. More cases need to be treated and reported using the bioabsorbable implant technique to fully assess its potential.

We did not record the lameness degree before or after surgery. Even though only 64.6% raced, the soundness prognosis after any of the procedures was reported as higher in past reports [16,20]. It is our perception that only a few horses are lame as yearlings because of these lesions, and SCLs are usually diagnosed incidentally in radiographic surveys. The current recommended treatments for yearlings with SCLs attempt to prevent future soundness issues, improve the radiographic appearance of the affected bones to increase a possible purchase value at an auction, and prevent expansion of the lesions [2,15]. In our study, a younger horse at the time of surgery was associated with an earlier first race. The prognosis after treatment of an SCL in the medial femoral condyle has been determined to be better in horses less than 3 years old [19]. It is unknown if this could be true for lesions at the third metacarpal/tarsals and phalanxes, as all literature-reported horses were of similar age at the time of surgery [20,21,22]. Even though clinical treatment success allows the horses to start racing and, in some cases, have radiographical resolution, the articular cartilage defect usually persists and might be a cause of shortened athletic careers due to the development of arthritis or kissing lesions on opposite cartilage surfaces [25]. It is unknown if horses that had not yet developed lameness at the time of treatment have a better prognosis than the ones that are already symptomatic; however, our objective was to report racing prognosis and not soundness. Further studies are needed to correlate lameness and racing prognosis in horses with SCLs.

The limitations of this study include those inherent in retrospective studies. Statistical results must be assessed with care, as some racing parameters have a high numerical range or high standard deviation, possibly requiring a greater number of horses to achieve more conclusive results. Racing earnings have been reported as a difficult parameter to measure racing performance due to the great differences in prizes for differently ranked horses and might not be statistically significant due to their high standard deviation [24]. Most racing parameters provide some information about racing quality or racing longevity, but none of them alone are perfect to reflect athletic outcomes [24]. The innate athletic ability of a horse varies between individuals, and better innate athletic ability might not be distributed equally among groups. Both the control and case groups have horses with common osteochondral disease lesions that may have affected performance. Errors using the radiographic imaging viewing software measurement tool could exist, and some superimposition of the lesions with the sesamoid bones can make the limit of the radiographic lesion difficult to clearly detect. To minimize the measurement errors, each lesion was measured by three different observers with experience in radiographic interpretation and used to acquire the radiographic measurements.

## 5. Conclusions

Subchondral cystic lesions in the phalanxes and third metacarpal/tarsal bones in yearling thoroughbred racehorse prospects negatively impacted their auction sale values but not their ability to start a race. SCLs of greater height were associated with decreased racing performances and a lower number of winning races. SCLs in the medial aspect of the joints were associated with decreased odds of starting a racing career. Treatments for SCLs with either corticosteroid injection, transcortical debridement, cortical screw, or bioabsorbable implant placement had similar racing performance prognoses.

## Figures and Tables

**Figure 1 animals-13-02838-f001:**
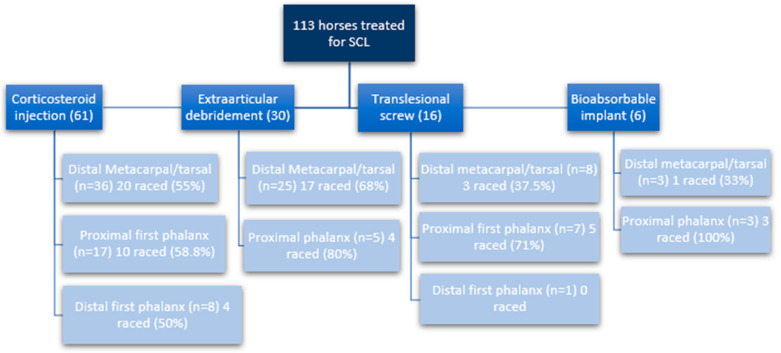
Flowchart of cases and their ability to start at least one race.

**Table 1 animals-13-02838-t001:** Radiographic appearance and shape classification into 3 types of subchondral lucencies based on their shape and communication with the joint.

Type 1	Subchondral lucency with a dome shape and an articular surface defect of similar width as the middle of the lesion.	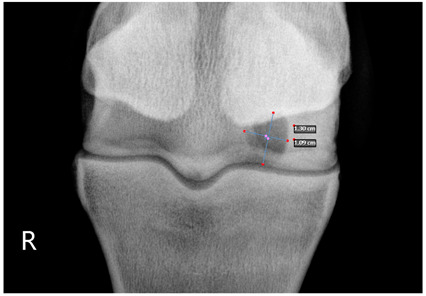
Type 2	Subchondral lucency with narrow communication with the joint surface (narrow cloaca) and round, wider lucency deeper in the bone.	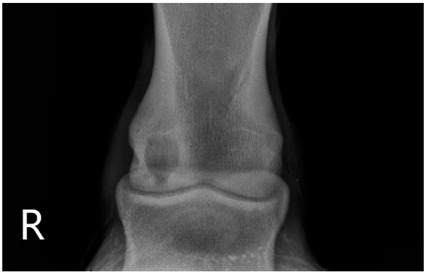
Type 3	Subchondral lucency with no radiographic evidence of communication with the joint surface.	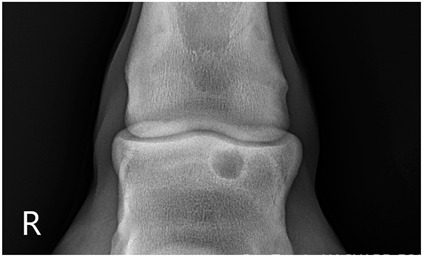

**Table 2 animals-13-02838-t002:** Distribution of SCLs by grades based on location and organized by type of treatment.

	Distal Metacarpal/Metatarsal			First Phalanx			Second Phalanx		
	Type 1	Type 2	Type 3	Type 1	Type 2	Type 3	Type 1	Type 2	Type 3
Intralesional injection with corticosteroids	18 (29.5%)	13 (21.3%)	5 (8.2%)	5 (8.2%)	8 (13.1%)	4 (6.5%)	0	3 (5%)	5 (8.2%)
Transcortical drilling	14 (46.6%)	9 (30%)	2 (6.7%)	1 (0.3%)	2 (6.7%)	2 (6.7%)	0	0	0
Cortical screw placement	4 (25%)	3 (18.7%)	1 (6.2%)	0	6 (37.5%)	1 (6.2%)	0	0	1 (6.2%)
Composite bioabsorbable implant	1 (16.7%)	2 (33.3%)	0	0	2 (33.3%)	1 (16.7%)	0	0	0

**Table 3 animals-13-02838-t003:** Performance results for horses grouped by type of treatment received for an SCL.

Group	N (Sex Distribution)	Number of Horses that Raced (%)	Median Total Earnings in US Dollars (IQR)	Median Wins (IQR)	Median Placed (IQR)	Mean Age at Surgery in Days (95% CI)	Mean Age at the First Race in Days (95% CI)
Intralesional injection with corticosteroids	61 (33 females/28 males)	34 (55.7%)	40,095 (4245–84,829)	2 (0–3)	2 (0–4.5)	378 (346–409)	1060 (984–1136)
Transcortical drilling	30 (13 females/17 males)	21 (70%)	46,588 (11,185–97,255)	1 (0–3)	2 (1–4)	380 (331–429)	1026 (954–1098)
Cortical screw placement	16 (5 females/11 males)	8 (50%)	50,660 (26,917–67,276)	2.5 (1–3)	4 (2–6.75)	324 (272–376)	1037 (893–1182)
Composite bioabsorbable implant	6 (3 females/3 males)	4 (66.6%)	61,059 (15,337–126,613)	2 (0.25–3)	4 (1–7.75)	417 (312–521)	1031 (861–1200)

**Table 4 animals-13-02838-t004:** Comparison of racing performance and auction results between cases and controls. * significant difference at *p* < 0.01.

	Median Sell Price in US Dollars (IQR)	Median Total Earnings in US Dollars (IQR)	Median Starts (IQR)	Median Wins (IQR)	Median Placed ([IQR] Range)	Mean Age at the First Race in Days (95% CI)
Controls	70,000 * (21,347–100,000)	54,448 (17,183–92,067)	10 (5.5–20)	2 (1–3)	3 (1–4.75)	1023 (986–1059)
SCL	37,000 * (15,000–60,000)	33,994 (11,824–81,331)	8 (5–19)	1 (0–3)	2 (1–5)	1045 (996–1094)

## Data Availability

The data presented in this study are available on request from the corresponding author. The data are not publicly available due to privacy and ethical restrictions.

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
