# Peer review of "Treatments and Prognosis for Subchondral Cystic Lesions in the Distal Extremities in Thoroughbred Prospect Racehorses"

_animals, 2023, doi:10.3390/ani13182838_

Round 1
Reviewer 1 Report
Simple summary
Line 14: as there is a lot of doubt about the etiology in the literature, please note that fact.
Abstract:
Line 35: "selected" racing parameters
Line 37: Please reword as "similar chances to race" is unclear. Try something like: There was no difference in the ability to start a race between cases and controls.
Line 43: Last sentence is awkward. Perhaps: Treatment did not influence the ability to start a race
Introduction
Line 49: Awkward, leave off the beginning modifying phrase.
Line 55: you sure ref 4 said this?
Line 59-61: Reference 5 is a mechanics paper. It suggests mechanics may affect bone resorption. Inflammation contributes, so suggest you stick with Steroid injection people and von Rechenberg.
Line 59- So it’s not really a "Core" it’s more fibroblasts detected histologically in soft tissue
Line 75: Best? How decided, highest number? is it statistically higher? And how do you resolve unimproved racing results with the "best" soundness? Maybe its meaningless or biased?
Line 82: not a few reports: one report of one horse.
M&M:
Line 103: not sure what this sentence means
How were treatments selected? Did any character of the lesion impact method chosen? Clinician preference, Cost of procedure, anything?
I see you gave addressed it in the discussion, but it belongs in M&M or results
Results:
Line 195: where is the SCL metric data? It's impact on racing is one of you major conclusions. You also mention correcting for magnification, but it’s not well described, perhaps a radiograph with an example would help.
Figure 1 should be checked for consistency between boxes. What does (n+7) mean?
Table 2 is difficult to understand and should be reformatted. its also cut up in the PDF, making that worse. Might do better in landscape orientation.
Lines 200-211 This paragraph should be placed after the first one in results
Discussion
Line 222-223: I'm confused. I thought there was no statistical difference in racing metrics between Cases and controls
Lines 233-234: Still confused. Does SCL morphology impact performance, or no?
Lines 238-242: Please provide this data and let’s have a statistician decide if your analysis is correct.
Line 231: Be cautious with the references. For instance, ref 7 has nothing to do with outcomes.
Line 251: This paragraph is wholly speculative and misleading. It is fine (even desirable) for the authors to provide context and for their results and propose their opinion on the project, but bias and speculation should be minimised..
Line: 251- This sentence comes out of nowhere, and refutes the one before it, as you say there was no difference in success, and then speculate without any support about the mechanisms of why some procedures are superior. And what constitutes "complete" removal: how do you know? And screw placement is surgical but does not remove the fibroblasts, check lines 251-252
Line 255: Please check your references. Specifically, ref 5 has nothing to do with the statements in this sentence. You should reference von Rechenberg, and the most she ever said was that strong signals for IL-6 were found in the fibroblasts (your ref 6) and that they "may" along with endothelial cells be responsible to produce local mediators (Vet Surg 2000;29:420-429.)
Line 261: Did you look at radiographic improvement? This sentence reeks of bias.
Line 264: "Soundness" determined by the vet who referred the horse for the technique is weak evidence. Also [16] treated SCL in several locations, so comparing it to MFC may not be wise. Using the success metric for MFC used by most publications on racing horses, their result was 73%, and about a third were Standardbreds, which may have a different outcome to TBs.
Line 275: This information belongs in results
Line 279-281: Not sure what this sentence means. But it has an interesting figure not reported elsewhere: 60% racing, which is on the low end of previously reported rates for the MFC and similar to doing nothing. Can you discuss why? And could your numbers be affected by horses still racing?
Line 285-288: Not sure what this sentence means, if it's about age, did you analyze that? Was it different between techniques? And what is the age range you used for "yearling"? It looks like treated horses ranged from 9-17 mos., which is quite wide.
Overall: Information is good. Recheck references for correct citations. Be more cautious about speculation in discussion.
/
Author Response
We really appreciate the very helpful comments and corrections. Please find a point by point disclosure of which action was done to address each concern.
Line 14: as there is a lot of doubt about the etiology in the literature, please note that fact.
Fact included in abstract
Line 35: "selected" racing parameters
Word included as recommended
Line 37: Please reword as "similar chances to race" is unclear. Try something like: There was no difference in the ability to start a race between cases and controls.
Sentence reworded
Line 43: Last sentence is awkward. Perhaps: Treatment did not influence the ability to start a race
Sentence reworded
Line 49: Awkward, leave off the beginning modifying phrase.
Sentence corrected.
Line 55: you sure ref 4 said this?
Reviewer was right. It does compare narrow communication vs dome shape but just expected changes in treatments. And says that different prognosis can be hypothesized based on depth but do not mention width at articular communication. Sentence was changed to include shape and size as a possible factor in treatment result with respective references.
Line 59-61: Reference 5 is a mechanics paper. It suggests mechanics may affect bone resorption. Inflammation contributes, so suggest you stick with Steroid injection people and von Rechenberg.
Yes, maybe a change in the order of references may have cause this error and some others in the manuscript. Reference was changed.
Line 59- So it’s not really a "Core" it’s more fibroblasts detected histologically in soft tissue
Changed the word core for fibrous center rich in fibroblast and endothelial cells.
Line 75: Best? How decided, highest number? is it statistically higher? And how do you resolve unimproved racing results with the "best" soundness? Maybe its meaningless or biased?
Changed the overstated word best for highest percentage reported.
Line 82: not a few reports: one report of one horse.
Changed to one
Line 103: not sure what this sentence means
Sentence erased
How were treatments selected? Did any character of the lesion impact method chosen? Clinician preference, Cost of procedure, anything? I see you gave addressed it in the discussion, but it belongs in M&M or results
Sentence added in methods to clarify case allocation to kind of treatment
Line 195: where is the SCL metric data? It's impact on racing is one of you major conclusions. You also mention correcting for magnification, but it’s not well described, perhaps a radiograph with an example would help.
Metric data included in the results divided by location of the lesion
Figure 1 should be checked for consistency between boxes. What does (n+7) mean?
Figure corrected
Table 2 is difficult to understand and should be reformatted. its also cut up in the PDF, making that worse. Might do better in landscape orientation.
Landscape orientation was adapted. attached is a table in just one page for better interpretation, as the one in the text cuts off again.
Lines 200-211 This paragraph should be placed after the first one in results
Paragraph moved
Line 222-223: I'm confused. I thought there was no statistical difference in racing metrics between Cases and controls
Sentence corrected
Lines 233-234: Still confused. Does SCL morphology impact performance, or no?
Changed word morphology for shape in all the document for consistency and clarity
Lines 238-242: Please provide this data and let’s have a statistician decide if your analysis is correct.
This result was overstated and sentence was changed. Statistician was used for previous and current changes in analyses.
Line 231: Be cautious with the references. For instance, ref 7 has nothing to do with outcomes.
Changed. Bibliography reviewed.
Line 251: This paragraph is wholly speculative and misleading. It is fine (even desirable) for the authors to provide context and for their results and propose their opinion on the project, but bias and speculation should be minimised..
Paragraph rephrased to minimize speculation.
Line: 251- This sentence comes out of nowhere, and refutes the one before it, as you say there was no difference in success, and then speculate without any support about the mechanisms of why some procedures are superior. And what constitutes "complete" removal: how do you know? And screw placement is surgical but does not remove the fibroblasts, check lines 251-252
Changed as in pervious comment
Line 255: Please check your references. Specifically, ref 5 has nothing to do with the statements in this sentence. You should reference von Rechenberg, and the most she ever said was that strong signals for IL-6 were found in the fibroblasts (your ref 6) and that they "may" along with endothelial cells be responsible to produce local mediators (Vet Surg 2000;29:420-429.)
Corrected
Line 261: Did you look at radiographic improvement? This sentence reeks of bias.
Sentence changed
Line 264: "Soundness" determined by the vet who referred the horse for the technique is weak evidence. Also [16] treated SCL in several locations, so comparing it to MFC may not be wise. Using the success metric for MFC used by most publications on racing horses, their result was 73%, and about a third were Standardbreds, which may have a different outcome to TBs.
I know I wanted to criticize this very biased [16] publication and came out the opposite. Changed the context of the paragraph.
Line 275: This information belongs in results
Paragraph moved to results.
Line 279-281: Not sure what this sentence means. But it has an interesting figure not reported elsewhere: 60% racing, which is on the low end of previously reported rates for the MFC and similar to doing nothing. Can you discuss why? And could your numbers be affected by horses still racing?
It seems not affected by horses that still hadn’t race as none of the horses that were not racing before started after this year and by contact with the farms they will not race in the future.
Maybe MFC has better prognosis for racehorse, however they do not use controls to compare how their unaffected population raced.
It is very speculative, but probably doing nothing might have the same outcome that any of the treatments but nobody has done any randomized study using a placebo group and probably nobody will do.
Line 285-288: Not sure what this sentence means, if it's about age, did you analyze that? Was it different between techniques? And what is the age range you used for "yearling"? It looks like treated horses ranged from 9-17 mos., which is quite wide.
Sentence changed. We did analyze age and results were that youger age at surgery yourger age at first race. We consider yearling less than 24 months, however nobody has discussed when a weanling (aprox 5 mo old) becomes a yearling.
A sentence was added in the methodology to clarify what we considered a yearling.

Reviewer 2 Report
Animals-2494108 Treatments for distal extremity subchondral cystic lesions and prognosis in Thoroughbred prospect racehorses
Summary: This is a retrospective study in which the authors report the outcome of 113 Thoroughbred horses treated for subchondral cystic lesions (SCL) of the distal limbs. Unaffected maternal siblings are used as a control group for comparison of performance results. No difference in outcome is reported between four treatment approaches, and affected individuals were reported to perform equivalently to their siblings. SCL size was correlated with some performance measures.
General Comments: This paper is interesting in that it addresses prognosis after treatment for SCL of the distal limbs, which are underrepresented in the existing literature. However, the study design and analytical approach have major flaws, there are missing data, and the authors overstate their conclusions (correlation is not causation!). Major flaws include the unbalanced nature of the groups being studied (including several groups with fewer than 5 individuals and one with n=1), the fact that treatments were not randomized, and the number of years across which the cohort was gathered (affecting number of years eligible to race). If this were simply a descriptive study, then these would not be as problematic, but the fact that the authors are attempting to do statistical analyses makes them hugely important. Therefore, I would strongly recommend considering reframing this paper as a descriptive study, in which comparison of the treated cohort with their unaffected siblings would be appropriate. If attempting prediction/association is strongly desired (despite the inherent design flaws) then a more appropriate statistical approach would be multivariable regression analysis, where multiple factors could be taken into account simultaneously to determine which, if any, are predictive of outcome. Predictive variables could include size (or category) of lesion, location of lesion, treatment type, sex, age at surgery, etc. Outcome variables could include whether or not the horse started a race (binary outcome, logistic regression), how many starts/wins/places (zero-inflated count outcome, negative binomial regression), earnings (continuous outcome, generalized linear regression), etc. Consultation with a statistician is recommended if the authors are not familiar with these approaches. While the bulk of the paper is written in appropriate English, there are several instances of odd syntax and inappropriate word choice that must be corrected. Recommend careful reading and editing by a native English speaker familiar who is familiar with the scientific topic and is a strong writer themselves. Specific comments are below.
Summary/Abstract
Line 15: SCLs are not classified as OCD. If caused by a failure of endochondral ossification then they could be considered a form of osteochondrosis (OC). They are also correctly classified as a developmental orthopedic disease (DOD) or juvenile osteochondral condition (JOCC).
Line 17-18: This sentence is awkwardly phrased and should be revised.
Line 19: You cannot improve the inherent athletic ability of a horse with SCL. You are trying to improve the prognosis for athletic use.
Line 19: You are not really comparing 4 surgical treatments – intralesional steroid injection is not a surgery. Better to just say “four treatments”. This needs to be fixed throughout the manuscript.
Line 43-44: This is not correct English grammar. Please revise.
Line 45: OCD is not an appropriate keyword. You could use osteochondrosis, developmental orthopedic disease, or juvenile osteochondral condition.
Introduction
Lines 49-50: This sentence (“When attending…”) does not make sense as written. Please revise.
Lines 85-86: Again, awkward phrasing that requires revision. Also, please check tense throughout this last paragraph.
Methods
Line 103: What were your exclusion criteria that you “did not find”? This sentence needs to be re-written in proper English.
Lines 105-107: How did you account for the fact that you had horses ranging in age from 2-8 years old at the time of your performance results data cut-off? Older horses have had more opportunities for starts and earnings. Since your median age to return to racing was 3 years of age, then that suggests that several of your horses that will return to racing have not yet had the opportunity to do so. This may skew your results.
Line 114: Here you refer to cyst depth, elsewhere it is referred to as cyst height. Please be consistent with your terminology throughout the manuscript.
Line 117: What was your inter-individual variability.
Lines 124-144: How were treatments assigned? If not random, then a comparison isn’t really fair. For example, if only Grade 1 lesions could be injected under ultrasound guidance, then it’s not really a comparison between treatments because they are being used for different things.
Line 151: Animals do not have gender (mental identification with a sex); this should read sex. Please correct here and elsewhere in the manuscript.
Results:
Line 165: The word “posterior” does not make sense in this context and should be replaced with “following”.
Figure 1: This figure needs to be proofed (+ instead of = in a couple of places). It also illustrates your massively imbalanced treatment groups. The maternal siblings don’t really belong in this figure. Also, what happened to your lesion grades? You never report them in the results section. A table showing the distribution by grade and bone location, organized by treatment group, is missing.
Table 2: Given the data distribution, you should be reporting median, interquartile range, and range for each of these variables.
Table 3: Given the data distribution, you should be reporting median, interquartile range, and range for each of these variables. Also, how many seasons did these horses compete? Some were only 2 years old at the time of your cutoff while others were up to 8 years old.
Lines 184: Should read “sale price”, not “price value”. Also, you need to identify the prices presented as means or medians. They should be medians and should be presented with their interquartile range +/- complete range.
Line 193: You had massively imbalanced groups and were likely underpowered to detect any differences.
Lines 195-195: You ran a correlation test, not an association test. Also, you cannot just report the p-value. You have to actually report the correlation and a 95% confidence interval, in addition to the p-value.
Discussion:
Line 214: Were most of these horses sold before treatment? I.e. were all these lesions visible on presale radiographs?
Line 216: Here and elsewhere you are overstating your findings. You have a correlation (the strength of which I can’t judge since you didn’t report it), not association, and certainly not causation.
Line 223: As above, this is a correlation.
Line 225-226: How would you make a determination about which horse/type of lesion should have earlier intervention? What is the evidence for spontaneous healing of these lesions as is seen with articular osteochondrosis?
Lines 233-234: These data are not reported in the Results.
Line 238: Again, only a correlation, which you do not actually report.
Line 246: You also failed to account for the wide range in the number of years your cohort was eligible to race.
Lines 248-249: This statement is difficult to believe and certainly impossible to prove. Suggest removing as it does not contribute to your conclusions.
Lines 275-279: This paragraph speaks to the treatment biases in your cohort – this is a major flaw.
Line 282, Lines 292-293: Improper English, does not make sense as written and needs to be revised.
Line 300: You do not report on racing longevity, although it is certainly a very important outcome.
Conclusions will need to be revised based upon revised data reporting and analysis.
While the bulk of the paper is written in appropriate English, there are several instances of odd syntax and inappropriate word choice that must be corrected. Recommend careful reading and editing by a native English speaker familiar who is familiar with the scientific topic and is a strong writer themselves.
Author Response
We appreciate all the comments and corrections. Please find a word document attached with response to all comments. We hope we have fulfilled your request and expectations for this manuscript. Please let us know if you find room for any additional improvement.
Thank you very much for your time.

Reviewer 3 Report
The aim of the manuscript is to determine differential race performance prognosis in Thoroughbreds undertaking surgical treatments for subchondral cystic lesion (SCL) of the distal limb by comparing 4 surgical treatment methods. The manuscript demonstrated that all four treatment options had similar racing performance outcomes, and that a much wider SCL in the distal limb can impact on race wins and places. The results will be useful in decision making during Thoroughbred sales and consideration of surgical treatment of distal limb SCL.
The manuscript is well structured and concise with citation of references relevant to the topic. Study design and methodology for this study are appropriate in my opinion.
Specific comments are as follows:
|
Line number |
Comments |
|
Line 57 – 61 |
According to the study cited - von Rechenberg et al, 2001, upregulation of IL1b and IL6 mRNA expression may be responsible for expansion of SCL. Is there any reason IL6 was mentioned but IL1b was not mentioned in this manuscript? To the reviewer’s knowledge, both IL1b and Il6 may be responsible for persistence, or expansion, of the lesion. |
|
|
|
|
Table 1 |
Row 1: …or larger width than the top of the SCL. Only the radiograph in row two has label with the limb side. Can the authors do the same for the remaining radiographs? |
|
Materials and Method |
|
|
Line 146 – 159 |
Day of first start and day of first race. Are these terms the same? Please be consistent or provide definitions if they are different. |
|
Line 165-166 |
Please provide the summary statistics for the control group here as done for the case group. |
|
Line 181 – 183 |
Were the horses with a lesion sold before or after the treatment? It is not clear in this manuscript when before or after the auction the treatment was performed. I can see from the discussion that the SCL were detected on sale radiographs however, it would be great to provide this piece of information in the materials and methods. |
|
Line 296 |
Whilst the authors have done well for selecting maternal siblings as control for the SCL case group, it can be argued that race performance may still be innately different between the groups. This is a limitation that should be stated in this manuscript in my opinion. |
|
|
|
A review of English Language is recommended.
Author Response
We really appreciate all the comments for making this manuscript better.
Please find attached a word document with the action taken to all the comments.
Thank you very much for your time.

Round 2
Reviewer 1 Report
Is improved, but still needs a little work. See comments.
Line 34: Do we know the siblings were free of SCLs? Were they free of any other performance limiting radiographic finding? It would be useful to better describe the control group.
Line 51: "are" relatively uncommon
Line 86: Needs editing and is a weak sentence. Equine distal extremity SCLs are infrequently reported and there are no studies comparing...
Table 1: Not sure why this isn't a figure, but ok. Table legend is wordy. The radiographic appearance of distal extremity subchondral lucencies (should not be abbreviated). Then do groups
Subfigure 1: A better radiograph is needed. I can see a reasonably intact subchondral plate, so not clear about it being wider at the joint, and the height of the SCL (not plural) cannot be determined because the sesamoid is in the way. It also has significant motion. Would be useful to add a better image with margins you measure indicted.
Subfigure 3: Also, a radiograph with motion, get a better one. And the legend is wordy: try A subchondral lucency (should not be plural or abbreviated) without an obvious communication with the joint.
Line 150: cortical screws were placed across the SCL and absorbable implants were placed into the SCL.
Line 182: Suggest you avoid modifiers and state the data. ...(74.3%) and in the medial aspect in 71.1% of the joints (n=86).
Line 202: Mean age of first start (horses don't debut) and please give a number with a SD. Approximately doesn't cut it for continuous data.
Line 225: There was no difference (p=0.12) in the percentage of horses sold at yearling auction between cases (64.6%) and controls (73.4%). Continue this format with the next variables.
Line 239: Please provide the height data for subjects and controls.
Line 251: what's wined races?
Line 256-260: Is there statistics on this comparison? And the whole sentence is speculation.
Line 258: Publication(s)?
Line 262-264: ...yearling to start a race was not affected by the presence of SCLs or applied treatment.
Line 265: there are many references reporting this fact and they should be added.
Line 274: Suggest: This classification was modified from previous models (needs reference) and was based on the shape of the SCL at the articular surface. A larger articular communication (Sandler ref) has been suggested (as it is just an abstract) to reduce the ability to start a race by increasing the detrimental effects on the cartilage (could reference Hendrix on the MFC).
Line 278-280: Difficult to understand
Line 282-283: Associated? statistically? And now its width too?
Line 313: If you didn’t look at it, you cannot say it. End sentence at resolution.
There are some spelling and language issues, that need some cleanup.
Author Response
Thank you again for your time and dedication.
Please find attached a word document addressing the specific comments.

Reviewer 2 Report
Animals-2494108.R1 Treatments for distal extremity subchondral cystic lesions and prognosis in Thoroughbred prospect racehorses
Summary: This is a retrospective study in which the authors report the outcome of 113 Thoroughbred horses treated for subchondral cystic lesions (SCL) of the distal limbs. Unaffected maternal siblings are used as a control group for comparison of performance results. No difference in outcome is reported between four treatment approaches, and affected individuals were reported to perform equivalently to their siblings. SCL size and location was associated with some performance measures.
General Comments: The authors have chosen to maintain the attempt at statistical analyses rather than just presenting a descriptive study. Unfortunately a lack of understanding of the statistical tests that were used is apparent throughout the presentation of the results and the resulting conclusions. For example, regression analyses do not give you correlations (they give you associations between predictor and outcome variables in the form of linear coefficients, odds ratios, etc.). Since information about the specific regression models used for each outcome were not presented, I cannot comment on the appropriateness of those analyses. The major flaws in the study design remain, namely the unbalanced nature of the groups, lack of randomization of treatments, and number of years across which the cohort was gathered. The last flaw is the most easily addressed by only considering a set number of years of racing for each individual (i.e. performance through 4 years of age) rather than trying to report “lifetime” information without accounting for the number of years raced. The fact that different surgeons (number not reported) performed only one kind of treatment (on an unknown number of cases) does not negate the issue of lack of randomization – if anything, you have now introduced another variable (surgeon) into the mix that could affect outcome. Specific comments are below.
Summary/Abstract
Lines 14-15: This would be better phrased “SCL are a form of developmental bone disease, and while their etiology is incompletely understood, they may be a manifestation of osteochondrosis (OC).”
Lines 26-27: Awkward phrasing, should be revised.
Lines 39-40: You cannot just report p-values. You must have some kind of informative statistic.
Lines 41-44: You did not calculate correlations. Also throughout the paper, please remove contractions from sentences (e.g. “did not” rather than “didn’t”).
Introduction
Lines 58-59: This statement must be cited appropriately.
Line 89: Should read “and described in this report” (not “and studied”)
Lines 91-96: Verbs should be in past tense, not present tense. Please check throughout the manuscript for this error.
Line 97: Should read “fewer” not “lower”.
Methods
Line 131: Should read “regardless of” not “despite”. Also, here and elsewhere the word “prize” needs to read “price”.
Line 134: If surgeries were performed before auction, then auction price should not be listed above as a factor (not) considered when treatment was selected. It might be better to say “perceived value of the horse” rather than “auction price” in the statement above.
Lines 148-149: Awkward phrasing needs to be revised.
Line 155: Data WERE analyzed.
Lines 155-164: This section does not describe univariate analysis, which is when you look for an association between a single predictor variable and an outcome variable. This would more accurately fall under the heading of descriptive statistics, but I don’t think you need separate headings at all.
Line 166: What kind of regression analyses were performed? Count data (i.e. number of starts) and continuous data (i.e. earnings) require different statistical approaches and will have different results reported. This entire paragraph needs to be revisited.
Lines 168, 170: Regression analysis does not test for correlation. It tests for a predictive association between predictor variables and outcome variables.
Results
Lines 197-198: What do you mean by prognosis? What are you actually testing here? Ability to start a race? Something else? A p-value by itself means nothing. What was the statistical test used? If these data were not normally distributed as stated in the methods section, then you need to report median (IQR). Mean +/- SD could also be reported, but since this is not what you did your statistical comparison with, it is not appropriate to report by itself. This entire paragraph could be moved to a table.
Lines 204-205: You don’t describe any correlation tests. Is this supposed to be a regression coefficient that you are reporting? Technically, if you are using age in years at first start as your outcome, that is count data and is not appropriate for linear regression (needs Poisson or negative binomial).
Lines 211-215: I’m not really sure why this is in the results. If it is a description of what you did, then it goes in the methods. If it’s a justification for what you did, then it belongs in the discussion.
Table 3: This table needs to be reformatted such that it is actually readable. Also, IQR here and elsewhere is best reported as 2 numbers – the 25th percentile and the 75th percentile. Suggest rounding to the nearest whole number for all variables.
Lines 237-245: You need to actually include all of the regression analysis results as supplemental data. Also, as previously stated, these are not correlations. I’m not sure what you mean by “could not be significantly explained by the combination of all predictive variables.” If this is a reflection of a fit test for the regression analysis, then this needs to be explained further and certainly needs to be addressed in the discussion. For anything other than linear regression analysis, beta means nothing – it must be translated into an OR, IRR, etc. as appropriate for the specific regression you ran.
Discussion/Conclusion
Line 250: Should read “SCLs of greater height” not “higher SCLs”. This also occurs in other places in the manuscript and should be corrected throughout.
Lines 250-253: This is a very awkwardly phrased sentence that again makes the mistake of using the verb “correlated” when that is not what was tested.
Lines 256-257: Sales price as a yearling was not mentioned as a predictor variable in your methods, nor was it reported in results.
Line 267: You do not address treatment failure in your cohort. If you’re going to bring up this point here, then you should address it.
Lines 279-280: Awkwardly phrased sentence needs to be revised.
Line 283: You did not report anything about wider SCLs in your results.
Lines 288-289: What evidence do you have for this?
Lines 293-296: To actually address this concern/limitation, you need to know how many seasons each horse raced and either account for it or limit the number of years you consider as “lifetime”. Otherwise, your results for “lifetime” earnings, starts, etc. are not valid.
Lines 290-328: This entire section needs to be revised for relevance and readability.
Line 348: Should be “inherent” not “inherited”.
Line 363: Again, not a correlation and should read “SCLs of greater height”.
There remain notable errors in English usage/syntax/word choice and awkward phrasing in many locations (especially in the discussion) needs to be addressed. Some suggestions have been made in my comments. However, I recommend careful reading and editing by a native English speaker familiar who is familiar with the scientific topic and is a strong writer themselves.
Author Response
Thank you again for your time and dedication.
Please find attached a word document with the responses
I will try to attach the pdfs with the requested statistical results below
